# Are cross-sectional safety climate survey results in operating room staff associated with the surgical site infection rates in Swiss hospitals?

Yvonne Pfeiffer [ID],[1] Andrew Atkinson,[2] Judith Maag,[2,3] Michael A Lane,[4] David Schwappach [ID],[5] Jonas Marschall [ID] [2,3,6]

¹Research department, Stiftung fur Patientensicherheit, Zurich, Switzerland
²Department of Infectious Diseases, Inselspital University Hospital Bern, Bern, Switzerland
³Swissnoso, National Center for Infection Control, Bern, Switzerland
⁴Quality & Safety Operations, Parkland Health, Dallas, Texas, USA
⁵Institute of Social and Preventive Medicine, Universität Bern, Bern, Switzerland
⁶Division of Infectious Diseases, Washington University School of Medicine, St Louis, Missouri, USA

**Correspondence to**
Dr Yvonne Pfeiffer;
ypfeifferdrsc@gmail.com

## ABSTRACT

**Objectives** The aim of this study was to investigate the association between surgical site infections (SSIs), a major source of patient harm, and safety and teamwork climate. Prior research has been unclear regarding this relationship.

**Design** Based on the Swiss national SSI surveillance and a survey study assessing (a) safety climate and (b) teamwork climate, associations were analysed for three kinds of surgical procedures.

**Setting and participants** SSI surveillance data from 20 434 surgeries for hip and knee arthroplasty from 41 hospitals, 8321 for colorectal procedures from 28 hospitals and 4346 caesarean sections from 11 hospitals and survey responses from Swiss operating room personnel (N=2769) in 54 acute care hospitals.

**Primary and secondary outcomes** The primary endpoint of the study was the 30-day (all types) or 1-year (knee/hip with implants) National Healthcare Safety Network-adjusted SSI rate. Its association with climate level and strength was investigated in regression analyses, accounting for respondents' professional background, managerial role and hospital size as confounding factors.

**Results** Plotting climate levels against infection rates revealed a general trend with SSI rate decreasing as the safety climate increased, but none of the associations were significant (5% level). Linear models for hip and knee arthroplasties showed a negative association between SSI rate and climate perception (p=0.02). For climate strength, there were no consistent patterns, indicating that alignment of perceptions was not associated with lower infection rates. Being in a managerial role and being a physician (vs a nurse) had a positive effect on climate levels regarding SSI in hip and knee arthroplasties, whereas larger hospital size had a negative effect.

**Conclusions** This study suggests a possible negative correlation between climate level and SSI rate, while for climate strength, no associations were found. Future research should study safety climate more specifically related to infection prevention measures to establish clearer links.

## STRENGTHS AND LIMITATIONS OF THIS STUDY

⇒ This study used national surgical site infections surveillance data of high quality, also including infection events after discharge from the hospital.
⇒ For modelling purposes, safety climate and teamwork climate outcomes, as well as level and strength of climate perceptions, were differentiated.
⇒ Including three different types of surgical procedures allowed for controlling for influences stemming from inherent variations in risk profiles depending on the kind of surgery.
⇒ Potential under-reporting of surgical site infections may have had a biasing effect on the analysis.
⇒ Aggregating climate on the hospital level was necessary due to the ad hoc formation of surgery teams, but this could conceal certain trends on unit level.

the most common healthcare-associated infection, occurring in 2%–5% of inpatient surgeries and represent an economic burden to the healthcare system.[1] SSI rates may be reduced effectively by performing adequate infection prevention measures, such as the use of antibiotic prophylaxis or skin disinfection prior to incision. For example, the adherence to a standardised bundle of prevention measures reduced SSI after colon surgery in a retrospective cohort study, thus improving patient safety.[2] In order for healthcare workers to perform safety-relevant behaviours, safety climate is thought to play an important role. Specifically, compliance with infection prevention measures is thought to be influenced by the prevailing safety climate,[3 4] however, only with mixed supporting evidence so far.[5] Similarly, safety culture and climate have been used as an explanatory factor for failed implementation of preventive interventions that were successful in the original studies.[6]

'Safety climate' encompasses shared perceptions related to safety policies, procedures and

## INTRODUCTION

Healthcare-associated infections are a major concern for patient safety. Specifically, surgical site infections (SSIs) are considered

practices,[7] and is expected to guide the safety behaviour of workers. There is limited evidence from prior research indicating that safety climate may be linked to healthcare-associated infections, and SSI in particular: Fan *et al*[8] found that 9 of 12 safety climate dimensions were associated with SSI rates after colon surgery. To evaluate a programme reducing SSIs after colorectal surgery in Hawaiian hospitals, Lin *et al*[9] assessed safety climate at baseline and after implementation of the programme; however, no consistent patterns of association of (change in) SSI rate with (change in) safety climate dimensions could be identified.

In order to carefully evaluate safety climate as a characteristic of a unit or organisation,[10] climate *strength* needs to be assessed—in addition to the safety climate *level* indicating positive or negative climate evaluations: climate strength indicates how aligned the individual climate perceptions are with each other in a given unit and is considered a predictor for safety outcomes[11 12] that can provide valuable insights. While the level of safety climate scores may not differ or change over time, their strength, indicating consensus or divergence of perceptions regarding safety climate, may do so.[13 14]

Based on the broad coverage and longitudinal assessment of SSI captured by the Swiss surveillance system, this study relates SSI rates to safety climate scores. The study aims were to investigate (a) whether better safety climate *level* and higher safety climate *strengths* are associated with lower SSI rates, and (b) whether these relationships are also true for teamwork climate scores.

A prior study of our group showed that infection rates may vary with hospital size, meaning that larger hospitals have higher rates, confounded by their case mix and type of surgeries investigated.[15] Therefore, hospital size was considered in our analyses. That study also investigated whether quality of surveillance, assessed by external audits, has an influence on the measured SSI rate, and we considered 'quality of surveillance' as assessed with the audit score here, too: hospitals conducting a thorough surveillance demonstrate their ability to implement and follow relevant procedures, which we hypothesised may also be related to safety climate. Three types of procedures were included in the study: colorectal surgery, caesarean section and knee/hip arthroplasties. They were chosen for being the procedures for which the largest number of institutions report SSI rates in Switzerland.

As prior research exploring the relationship between safety climate and SSI rates has advocated[9] and for keeping potential intercorrelations between variables to a minimum, we used one composite safety climate scale. We added the scale 'teamwork climate' because teamwork is important for safe outcomes in surgeries, and it has been linked to healthcare-associated infections in the neonatal intensive care unit setting before.[16]

Prior research has also shown that those in managerial positions tend to rate safety climate higher than front-line healthcare workers,[17–19] and also that the professional background (here 'nurse' or 'physician') has an

influence on safety climate perceptions.[17 18] This has been traced back to various explanations, such as the existence of different management structures, subcultures or work demands in these two groups. Therefore, in this study, the managerial role and professional background were taken into account when analysing the relationship with infection rates.

## METHODS

### Sample and missing data

All hospitals and hospital networks providing acute care and participating in the SSI surveillance module of Swissnoso were invited (N=143) to participate in the study, of which N=54 institutions agreed. Among these, all operating room (OR) personnel received the survey. An exact response rate cannot be estimated, as total numbers of healthcare workers in ORs were not retrievable. However, we know that 38% of all paper surveys sent out were completed. There was a thorough description of the project's aims, that participation was voluntary and the use of the responses on the first page of the survey. Therefore, participation was considered informed consent.

After exclusion of participants with demographic answers that were inconsistent, for example, indicating that they worked as physician and as nurse at the same time (7 participants), and those with less than 32 (from 52) responses (36 participants), we had a sample of N=2769 (see online supplemental figure 1).

For SSI rates, we selected adult (≥16 years) patients with complete follow-up information in 2017–2019 (3 years) for elective hip and knee arthroplasties (58 495 procedures, 127 hospitals), colorectal surgeries (21 445 procedures, 126 hospitals) and caesarean sections (19 917 procedures, 49 hospitals) from the Swissnoso database (extraction performed on 30 September 2020). We assumed this would provide an adequate summary of the infection rates at the time of the survey in 2019. Combining survey data with infection rates meant limiting the sample size, as not all surgery types are reported by every hospital. Equally, for us to match survey responses with persons who work in the surveilled surgery types, each survey respondent had to list the type of surgeries they participate in. Data were collected on the patient and participant level, and then aggregated on hospital level. The resulting analysis set consisted of 20 434 surgeries for hip and knee arthroplasty from 41 hospitals, 8321 for colorectal procedures from 28 hospitals and 4346 caesarean sections from 11 hospitals (see online supplemental figure 1 and online supplemental table S1).

In addition to the complete case analysis, we multiply imputed missing responses from individual survey participants using predictive mean matching with the 10 nearest neighbours as donor pool in the algorithm.[20]

### Safety climate and teamwork climate measures

In order to assess climate, the safety climate (*SCS*, 22 items) and teamwork climate (*TWC*, 6 items) scales of the

Safety Attitudes Questionnaire (SAQ)[21] were used (see online supplemental appendix 1). The internal consistency of the *SCS* was high: Cronbach's alpha=0.89; for *TWC*, it was slightly lower with Cronbach's alpha=0.78, but still acceptably high.

The survey was sent out in the three national languages: a prior study had developed Swiss versions in German and French,[22] and the Italian version was developed by applying a translation and back-translation process using different translators, blinded to the original items. Any differences were resolved, and the items tested in two sites by OR personnel for understandability and correctness.

Two different measures were used in the analyses: safety and teamwork climate *level* and safety and teamwork climate *strength*. Climate level was measured as per cent positive responses (PPRs) per hospital. Percentage of positive respondents was calculated counting each respondent's score as positive for their unit if their mean score on the scale was 4.0 or greater (from a possible score of 1–5), in line with Tawfik *et al*'s approach.[14] Intrahospital correlation of the climate level scores (*rwg*, a measure of within-group agreement) was checked in order to justify aggregation on the hospital level: all *rwgs* were above 0.66, with their median being 0.78 (*SCS*) and 0.81 (*TWC*), respectively, indicating moderate to strong agreement.[23] In line with prior research, climate strength was evaluated using the SD of mean climate level scores per hospital.[10 14 24]

### Assessment and adjustment of SSI rates and incorporation of audit scores

Analyses were conducted separately for the surgery types, because SSI rates are not comparable between specialties since certain procedures are more prone to SSI than others. The primary endpoint of the study was the 30-day (all types) or 1-year (hip/knee with implants) National Healthcare Safety Network-adjusted rate of infection,[25] and this was assessed in all patients aged 16 years and older for the respective hospitals in the 3-year study period. For hip/knee surgeries, only class I ('clean') surgeries were included, but all classes were included for colorectal surgeries and caesarean sections.

Despite the sophisticated surveillance system employed in Switzerland, the accuracy of the SSI rates may be affected by under-reporting. Therefore, audit scores from on-site quality audits were also taken into account. These hospital audit scores indicate how thoroughly the SSI surveillance is conducted in a given hospital, with higher audit scores linked to higher rates of reported SSI for certain surgeries.[15] Audit scores range from 0 to 50, and cover aspects such as how structured the data assessment is, how well trained the persons conducting it are, among others.[26]

### Data analyses

The primary analysis was based on complete cases only and considered the correlation between the SSI rate, aggregated on the hospital level, and the safety climate and teamwork climate.

Categorical variables are shown with the number per category (N) along with the appropriate percentage of the total, with continuous variables summarised as median and IQR. Group differences were investigated using the Kruskal-Wallis non-parametric test.

We began by visually checking for associations by plotting safety and teamwork climate scores (level and strength) against the standardised infection rates for each procedure type, stratified by hospital size (<200, 200–499 and 500+ beds).

Weighted linear models were fitted with the respective safety climate endpoint as dependent variable, and as independent variables, the SSI rate, percentage of managerial (vs non-managerial) roles and percentage of physicians (vs nurses) having responded to the survey in a hospital for the specific surgery type, the hospital size and hospital type (university, regional or 'other'). The number of surgeries in a given hospital was used for the weighting in the models.

Those variables statistically significant at the 10% level in univariable analyses were included in the multivariable analysis with forwards selection and backwards deletion using the Akaike information criterion to determine the most parsimonious model.

### Patient and public involvement

To stay safe from SSI is important for patients undergoing surgery, and to study the role of safety climate thus can offer important avenues for improvement. From the time when the project was funded, the public was informed on the website of Swissnoso (National Center for Infection Control, Bern, Switzerland) and on the website and in the newsletter of the patient safety foundation. The patients were not involved in the design and conduct of the study.

### RESULTS

Mean safety climate level scores were very similar between the healthcare workers involved in the three procedure types (caesarean section 43.3, IQR (0.0–66.8), hip/knee median PPR 45.5 (12.5–83.3), colorectal 48.1 (4.0–100.0)), but as the ranges show, with large variability between hospitals. As expected, standardised infection rates mirrored the underlying population and surgery type differences between the surgery types (hip/knee median 1.0% (0.0–3.0), colorectal 16.0% (7.0–33.0), caesarean section 2.0% (1.0–4.0), top part of table 1; online supplemental figure 2). The median number of survey respondents per hospital was deemed adequate, although there were some imbalances due to low numbers in some hospitals. Approximately 20% of respondents had a managerial role and approximately 30% were physicians (as opposed to nurses), although the percentage of physicians was somewhat lower for caesarean sections. Interestingly, despite the comparable safety climate level scores, teamwork climate level scores indicated a slightly

**Table 1** NNIS-adjusted SSI rate compared with safety climate and teamwork scores, stratified by surgery type

| N (%)/median (IQR) | Hip and knee arthroplasty | Colorectal | Caesarean section |
|---|---|---|---|
| Safety climate (%) | 45.5 (12.5–83.3) | 48.1 (4.0–100.0) | 43.3 (0.0–66.7) |
| Number of hospitals | 41 | 28 | 11 |
| Number of surgeries | 20 434 | 8321 | 4346 |
| NNIS-adjusted infection rate | 1.0 (0.0–3.0) | 16.0 (7.0–33.0) | 2.0 (1.0–4.0) |
| Hospital size (beds) | | | |
| <200 | 27 (65.9) | 13 (46.4) | 8 (72.7) |
| 200–499 | 9 (22.0) | 9 (32.1) | 2 (18.2) |
| >500 | 5 (12.2) | 6 (21.4) | 1 (9.1) |
| Number of participants | 754 | 493 | 210 |
| Participants per hospital | 17 (5–53) | 12 (5–71) | 12 (6–40) |
| Percentage leaders | 20.0 (0.0–50.0) | 21.4 (0.0–80.0) | 18.2 (5.9–50.0) |
| Percentage physicians (vs nurses) | 30.0 (0.0–100.0) | 34.5 (0.0–100.0) | 20.0 (0.0–57.5) |
| Teamwork (%) | 48.1 (4.0–100.0) | 55.6 (7.4–88.9) | 54.3 (0.0–100.0) |
| Number of hospitals | 44 | 28 | 12 |
| Number of surgeries | 21 247 | 8321 | 4824 |
| NNIS-adjusted infection rate | 1.0 (0.0–3.0) | 16.0 (7.0–31.0) | 2.0 (1.0–4.0) |
| Hospital size (beds) | | | |
| <200 | 30 (68.2) | 13 (46.4) | 9 (75.0) |
| 200–499 | 9 (20.5) | 9 (32.1) | 2 (16.7) |
| >500 | 5 (11.4) | 6 (21.4) | 1 (8.3) |
| Number of participants | 861 | 549 | 239 |
| Participants per hospital | 18 (5–56) | 14 (5–72) | 14 (5–42) |
| Percentage leaders | 20.0 (0.0–50.0) | 21.4 (0.0–80.0) | 19.4 (5.6–44.4) |
| Percentage physicians | 33.3 (0.0–100.0) | 34.4 (0.0–100.0) | 22.0 (0.0–57.4) |

NNIS, National Nosocomial Infections Surveillance.

lower median value for hip/knee compared with the other two procedure types (48.1 compared with 55.6 and 54.3, respectively, bottom part of table 1).

Otherwise, the characteristics were very similar to those for the safety climate endpoint. The stratification by hospital size did not reveal any major differences, although these comparisons certainly lacked power in some of the subcategories (online supplemental figure 3). The SSI data used for the analysis were broadly comparable with that from the underlying Swissnoso population, with little discernible selection bias (online supplemental table S2).

Plots of the safety endpoints against the standardised SSI rate for each of the surgery types showed generally decreasing trends, in line with our hypothesis that as safety climate and teamwork increase, the SSI rate drops. However, with the exception of the safety climate endpoint for hip and knee surgeries (p=0.02 for slope in adjusted linear models, see table 2), these findings were not supported by formal statistical tests (figure 1). Similarly, strength of safety climate and teamwork climate, as measured by the SD of the respective metric, was also inconclusive, although once again safety climate showed a promising trend (figure 2).

Furthermore, univariable linear models indicated that higher percentages of both managerial roles and physicians were associated with higher teamwork scores for hip/knee and colorectal surgeries. For patient-level analyses, please see online supplemental table 3.

### Subgroup analysis

In a subgroup analysis of hip/knee surgeries, in which a marginal association with SSI rate was apparent, we investigated whether larger hospitals are, in some sense, more critical about their own work processes, and hence have lower safety climate level scores. The right-hand side of figure 3 confirms there is no difference in safety climate level scores based on hospital size (<200 beds, 200+ beds), despite a significantly higher SSI rate in larger hospitals (p=0.02). Factoring in the audit score, a measure of the SSI surveillance thoroughness, but here used as a proxy for 'adherence to safety-relevant protocols and standards', revealed a slightly lower median safety climate score in those with higher audit scores (ie, more thorough adherence) for larger (200+ beds) compared with smaller hospitals (figure 4, far-right set of box plots).

**Table 2** Estimates following fitting of unadjusted and adjusted weighted linear models

**A. Safety climate**

| Endpoint | Hip and knee | | Colorectal | | Caesarean section | |
|---|---|---|---|---|---|---|
| **% safety climate** | **Estimate, p value** | | **Estimate, p value** | | **Estimate, p value** | |
| | **Univariable** | **Multivariable** | **Univariable** | **Multivariable** | **Univariable** | **Multivariable** |
| SSI rate | −, 0.05 | −, 0.02 | NS | – | NS | – |
| % managerial role | +, 0.08 | NE | +, 0.01 | – | +, 0.1 | – |
| % physician | +, 0.08 | +, 0.03 | NS | – | NS | – |
| Hospital size | | NS | | – | Reference | – |
| <200 | Reference | | Reference | | NS | |
| 200–499 | NS | | NS | | NS | |
| 500+ | −, 0.09 | | NS | | | |
| Hospital type | | – | | – | Reference | – |
| Cantonal hospital | Reference | | Reference | | NS | |
| Others | NS | | NS | | NS | |
| University | NS | | NS | | | |

**B. Teamwork**

| Endpoint | Hip and knee | | Colorectal | | Caesarean section | |
|---|---|---|---|---|---|---|
| **% teamwork climate** | **Estimate, p value** | | **Estimate, p value** | | **Estimate, p value** | |
| | **Univariable** | **Multivariable** | **Univariable** | **Multivariable** | **Univariable** | **Multivariable** |
| SSI rate | NS | – | NS | – | NS | – |
| % managerial role | +, 0.1 | NE | +, 0.005 | – | +, 0.1 | – |
| % physician | +, 0.07 | +, 0.04 | +, 0.03 | – | NS | – |
| Hospital size | | | | – | | – |
| <200 | Reference | Reference | Reference | | Reference | |
| 200–499 | NS | NS | NS | | NS | |
| 500+ | | NS | NS | | NS | |
| Hospital type | | | | – | | – |
| Cantonal hospital | Reference | Reference | Reference | | Reference | |
| Others | NS | NS | NS | | NS | |
| University | −, 0.08 | −, 0.04 | NS | | NS | |

Since managerial role and physician are collinear, we include '% managerial role'; and hospital size and hospital type are collinear, we include hospital size.
+, increasing trend (ie, positively correlated); −, decreasing trend (ie, negatively correlated); NE, not estimated; NS, not significant at the 5% level; SSI, surgical site infection.

## Missing data analysis

We multiply imputed 20 complete data sets (N=2703 survey participants, compared with N=754 in the complete case analysis) making the missing at random assumption, including responses from all questions (safety climate and teamwork climate), along with the relevant covariates from table 1 in the imputation model, and then repeated the primary analysis for knee and hip arthroplasties. We compared descriptive statistics and the linear model estimates with those from the complete case analysis. Following multiple imputation, there was an increase in physician (and 'persons with managerial role') participation, indicating that physicians were less likely to complete the questionnaire (online supplemental table M1, for example, 30%–44% for physicians for the safety climate endpoint, p<0.001). Otherwise, the results were similar with slight, but not statistically significant, increases in safety climate and teamwork level score for the multiply imputed data (online supplemental table M2).

## DISCUSSION

The idea that a good safety climate in an institution is associated with good safety outcomes for its patients is prominent in health services research. By their incidence and susceptibility to prevention measures, SSI rates are a suitable safety outcome to be linked to measurements of safety climate. As the Swiss assessment of SSI is of high

### A. Knee and hip arthroplasty

#### i. Safety climate score

#### ii. Teamwork climate score

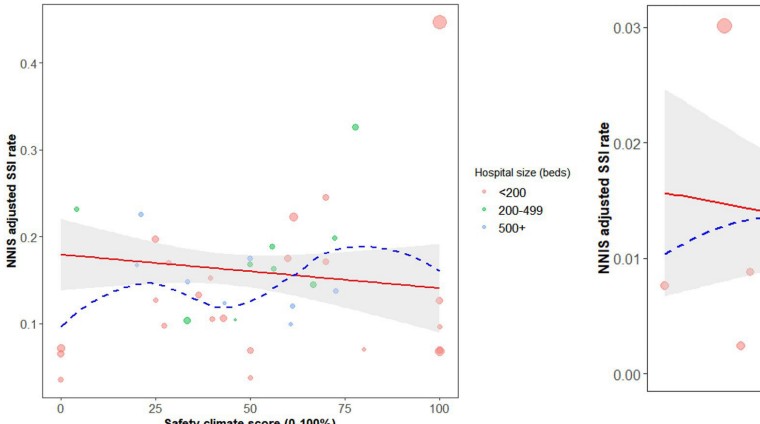
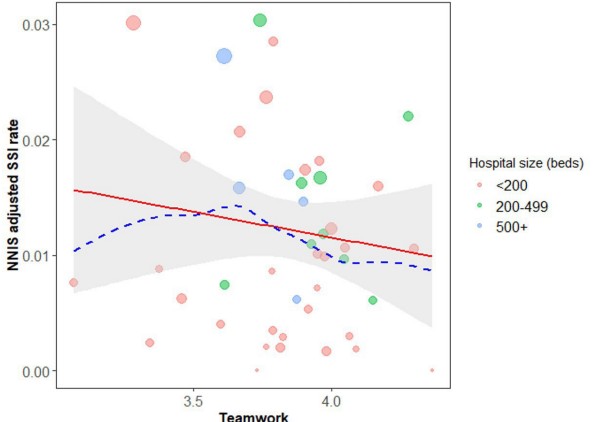

### B. Colorectal surgery

#### i. Safety climate score

#### ii. Teamwork climate score

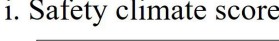
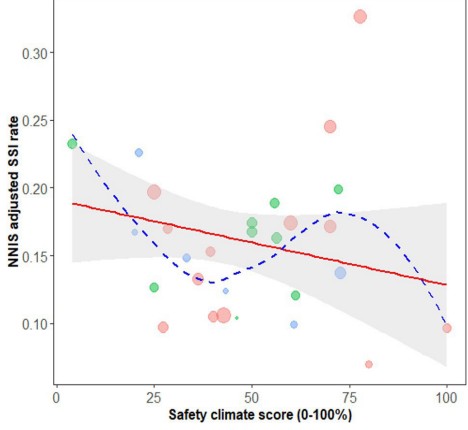
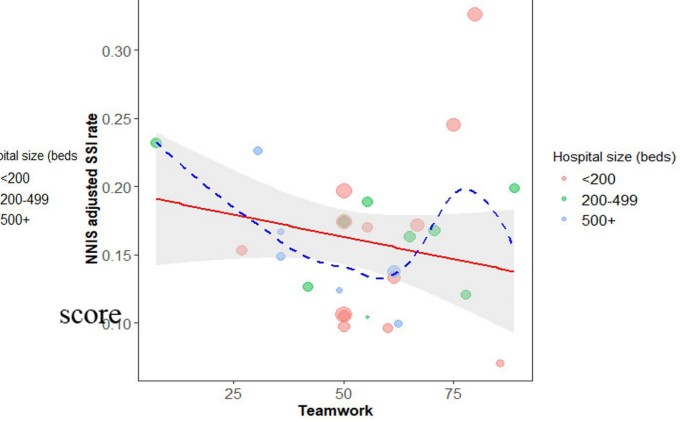

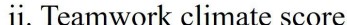

### C. Cesarean section

#### i. Safety climate score

#### ii. Teamwork climate score

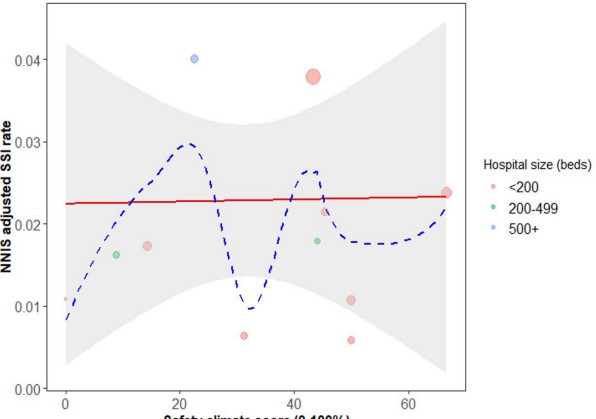

**Figure 1** Scores plotted against NHSN-adjusted infection rate: (A) hip and knee arthroplasty; (B) colorectal surgery; (C) caesarean section. Bubble sizes proportional to SE of the infection rate; weighted linear model shown (red solid) with 95% CI (grey shaded); LOESS (Locally estimated scatterplot smoothing) smoother added (blue dashed); weights are proportional to number of participants answering the safety climate survey in the respective hospital. NHSN, National Healthcare Safety Network; NNIS, National Nosocomial Infections Surveillance; SSI, surgical site infection

## A. Knee and hip arthroplasty

### i. Safety climate strength

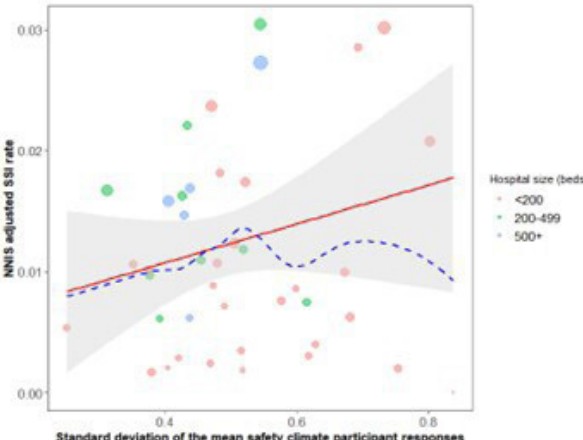

### ii. Teamwork climate strength

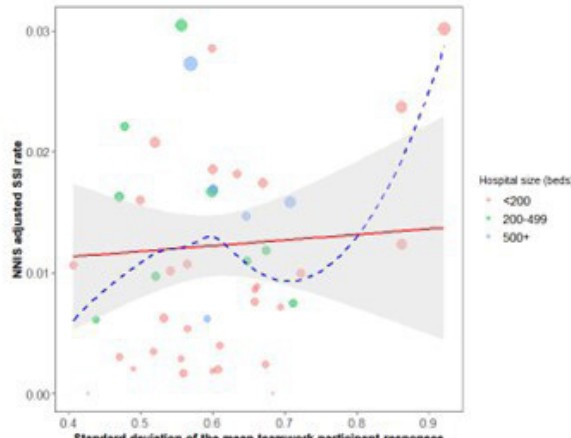

## B. Colorectal surgery

### i. Safety climate strength

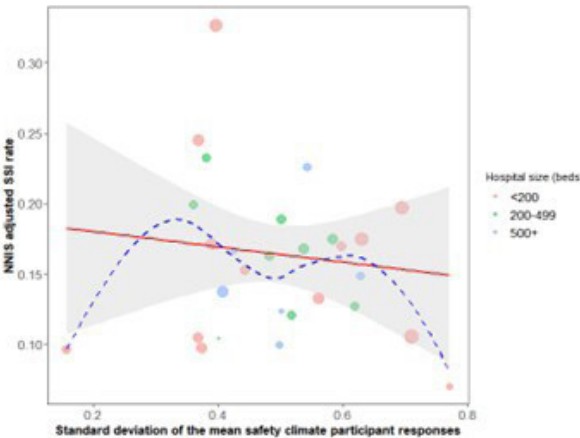

### ii. Teamwork climate strength

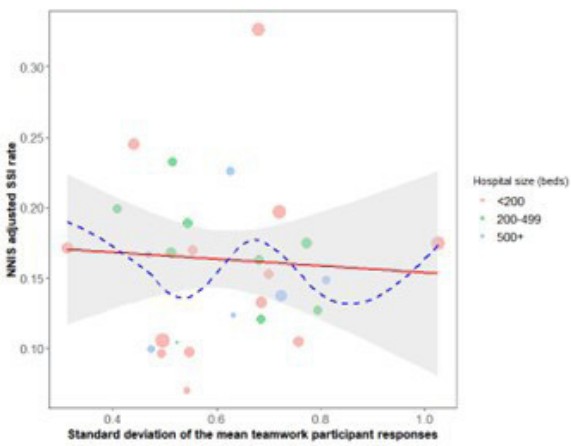

## C. Cesarean section

### i. Safety climate strength

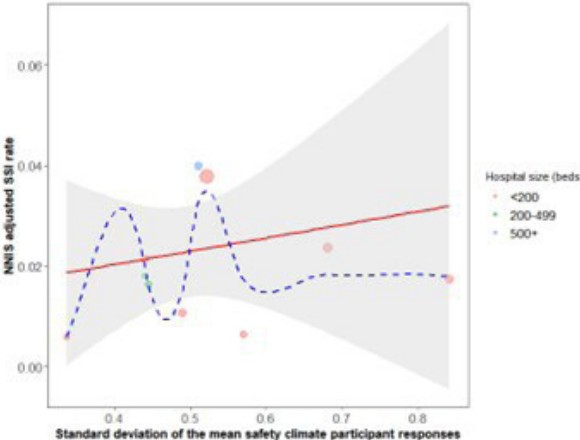

### ii. Teamwork climate strength

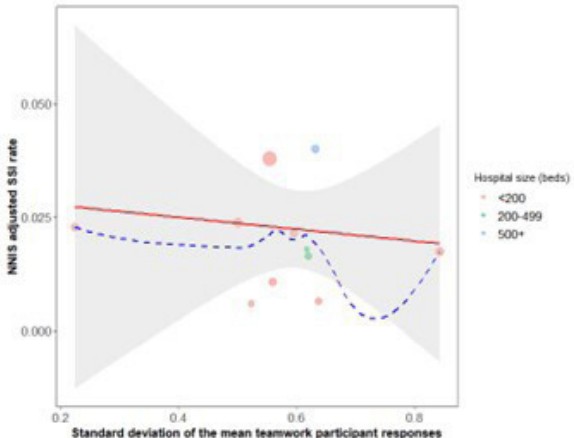

**Figure 2** SD of score plotted against NHSN-adjusted infection rate: (A) hip and knee arthroplasty; (B) colorectal surgery; (C) caesarean section. NHSN, National Healthcare Safety Network; NNIS, National Nosocomial Infections Surveillance; SSI, surgical site infection.

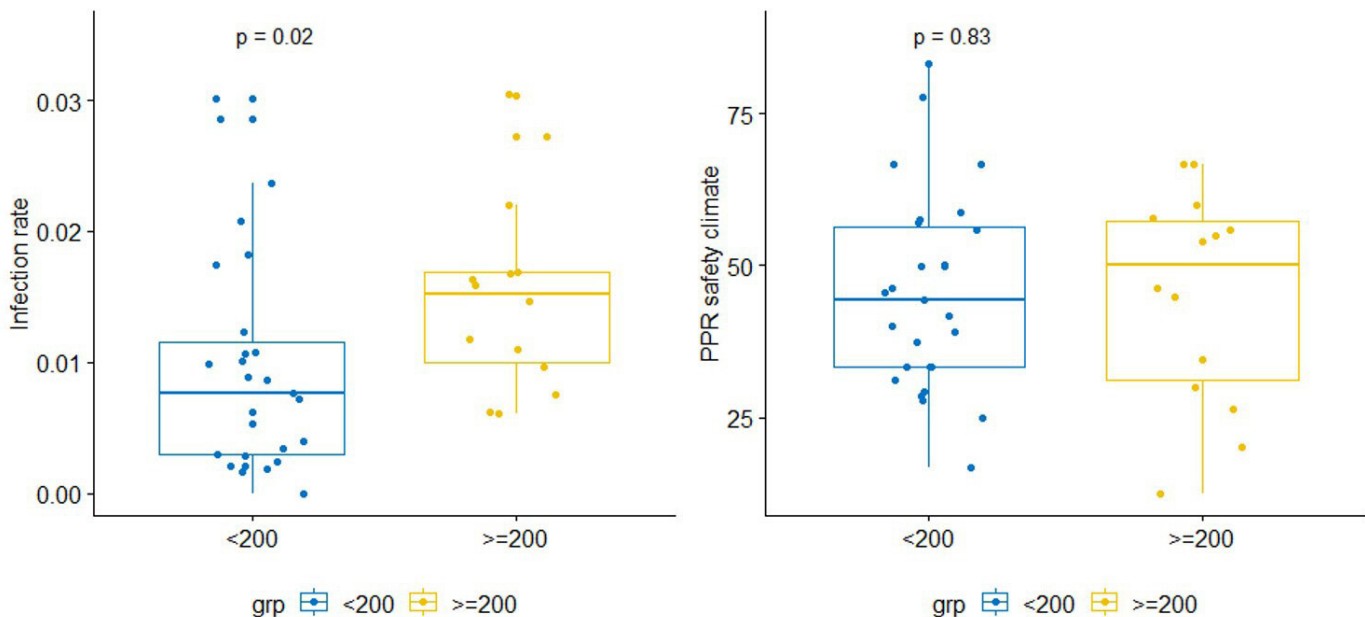

**Figure 3** Differences in SSI rate and safety climate according to hospital size. PPR, per cent positive response; SSI, surgical site infection.

quality (including infections becoming obvious after discharge from the hospital and the surveillance quality being audited on a biannual basis), this study represented a good opportunity to better understand the relationship between safety climate and safety outcomes. Indeed, this study revealed some associations between climate level and SSI rate in the expected direction, that is, the higher the safety climate, the lower the SSI rate. However, only for hip and knee surgeries was a formal statistical association observed. As implant surgery is 'cleaner' than the other surgery types, it is probable that the diligence of adhering to preventive measures is better reflected in the respective infection outcomes; this may explain why the association was most salient for these surgeries.

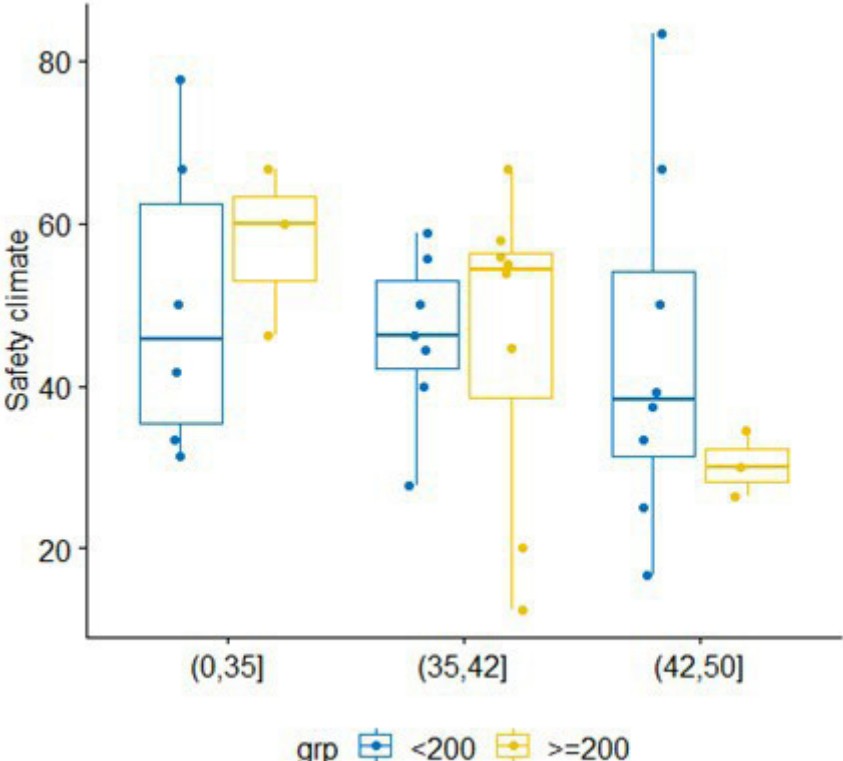

**Figure 4** Differences in safety climate ratings according to surgical site infection surveillance quality, stratified by hospital size.

Overall, however, this study did not empirically support the expected associations between safety and teamwork climate among Swiss OR personnel and SSI rates in a strong way. This result may be interpreted in two ways: that (a) safety climate is not as important as other factors that have not been assessed (such as knowledge of personnel or surgical technique used) or that (b) safety climate was not assessed specific and well enough to uncover existing strong associations.

Investigating the role of climate *strength* did not yield coherent and easily interpretable results, neither for safety climate nor for teamwork climate. We expected higher strength (ie, higher agreement of climate perceptions) to be reflected in lower SSI rates. When plotting the strength scores against the infection rates, this assumption was confirmed for knee and hip surgeries as well as for caesarean sections. However, for colorectal surgeries and for the relationship between teamwork climate and caesarean section SSI rates, the direction of the association was contrary to our expectation. As we evaluated climate strengths per hospital, these mixed findings might be traced back to the frequent ad hoc formation of surgical teams which makes it hard to define team boundaries and to develop aligned perceptions of climate.

To investigate whether larger organisations are more safety conscious and therefore have a more critical view of their own environment, we compared SSI rates and safety climate levels between larger and smaller hospitals for the arthroplastic surgery sample, but there was no difference in climate that could have corroborated this idea. In a further step, we explored whether the thoroughness of the SSI surveillance process was associated with safety climate for hip/knee surgeries (in which a marginal association with SSI rate was encountered): larger hospitals in fact tended to have lower safety climate scores, the higher their surveillance audit scores were. Here, we are interpreting a high audit score as a marker for the ability to comply with external regulations and to follow through in implementing procedures. A hospital's workforce that is capable of maintaining a high-quality surveillance and reporting process according to national requirements may therefore be less likely to report overly optimistic safety climate scores. Such an organisation obviously does much to adhere to standards and protocols, and staff may be enabled to provide a more realistic, or even critical, evaluation of teamwork and safety climate. The combination of low safety climate scores with high audit scores could therefore point to larger hospitals having a higher 'sensitivity to their operations', which is an important feature of high-reliability organisations.[27]

Having a managerial role or being a physician had a positive influence on climate levels for two surgery types. This result is in line with earlier research indicating that persons with managing roles have a more positive view on safety climate,[17–19] and with reports of physicians evaluating safety climate more optimistically than nurses,[28] although that difference is not as clear as for managerial roles.[29] Future studies will need to shed light on the origins of differences in safety climate perceptions and their potential effects on safety outcomes.

We decided to use the safety climate scale and the teamwork climate scale of the SAQ,[21 22] as the latter specifically covers nurse–physician collaboration in two items. Differentiating the composite safety climate dimension from the specific teamwork climate dimension revealed some nuances in the results, but overall, it did not provide particularly new insights. While Fan *et al*[8] in their prior study in colon surgery investigated different climate dimensions, Lin *et al*[9] argued for using composite scores, a suggestion that is supported by our results.

## Limitations

Climate could only be aggregated to hospital level, due to the usual ad hoc formation of surgery teams. Respondents' identification with a different group than with their hospital was not assessed and therefore some of the real-world complexity of climate perceptions may have been lost.

Both the number of patients and hospitals included in the study provide sufficient statistical power for investigating the trends in question. However, aggregation on the hospital level (and weighting) could conceal certain trends. However, a post-hoc patient-level analysis without hospital-level aggregation confirmed the results from the main analysis.

For the analyses following multiple imputation of missing data, the slope estimate when regressing safety climate on the SSI rate for hip/knee arthroplasties turned out to be statistically significant (p=0.003), which was not the case for the analysis including only the complete cases. We chose to include the more conservative estimates from the complete case analysis in the results. Therefore, we take the opportunity to highlight this potential bias, and the considerable effect of including more data from physicians in the analysis.

The assessment of the SSI rates may be biased due to under-reporting of infections and thus conceal certain existing associations. As the SSI rates also included an assessment of potential infections 30 days after the procedure in a telephone interview, this effect is minimised compared with studies assessing SSIs before discharge.

In adding to the inconsistent and in many cases nonsignificant[30 31] findings of the relationship of climate level and strength with various safety outcomes, such as medication errors,[32] patient evaluations,[31] length of stay[14] and healthcare-associated infections,[32] this study raises two areas for future research: first, safety climate surveys may not fully assess the perceptions that are actually relevant for performing specific safety behaviours such as adhering to SSI prevention measures. As Meeks *et al*[5] reported, for achieving compliance with SSI prevention measures, specific interventions were needed and having a good safety climate was 'simply not enough'. Groves[33] suspected that safety climate surveys only assess the declared values in an organisation, which is not necessarily what guides an individual's behaviours. For SSI

rates, this would mean integrating perceptions around performing the relevant SSI prevention measures such as administering the correct antibiotic prophylaxis in a timely fashion, for example. Applying additional qualitative approaches such as observations and interviews to learn more about the attitudes and perceptions around infection prevention measures could potentially be a fruitful avenue.[34] Second, to study the influence of safety climate on outcomes, it is necessary to make sure that the outcome is malleable to a large degree by safety-conscious behaviour. The fact that our results are clearer for arthroplastic surgery underscores this link. Thus, outcomes should be chosen that are highly sensitive to correct infection prevention behaviours and have as few confounding influences as possible.

## CONCLUSION

There is a growing body of research developed under the assumption that safety culture needs to be improved before or concurrently with the introduction of infection prevention measures or when a specific infection prevention measure is not showing success. However, this study adds to the inconsistent results around associations between safety climate and healthcare-associated infection rates. Future studies will need to apply other safety climate assessment methods, which should be tied more closely to the actual behaviour regarding infection prevention measures. Thus, using an outcome measure indicating whether prevention measures have been adhered to will be an important way to learn more about the link between safety climate and safety outcomes.

**Acknowledgements** The authors thank all nurses and physicians who responded to the survey and to the local coordinators facilitating the survey in the hospitals. They also appreciate Maria Mancuso (Clinica Luganese Moncucco) as well as Adriana Degiorgi and Céline Corti (both Ente Ospedaliero Cantonale) for facilitating the pretest of the Italian survey items. The authors thank all participating centres for providing their SSI surveillance data. These data were collected in collaboration with the Swiss National Association for the Development of Quality in Hospitals and Clinics (ANQ) and the National Center for Infection Prevention (Swissnoso). Members of the Watussi Study Group (in alphabetical order): Andrew Atkinson (Inselspital, Bern University Hospital and University of Bern), Lauro Damonti (Inselspital, Bern University Hospital and University of Bern), Philipp Jent (Inselspital, Bern University Hospital and University of Bern), Rüdiger Külpmann (Lucerne University of Applied Sciences and Arts), Judith Maag (Inselspital, Bern University Hospital and University of Bern), Jonas Marschall (Inselspital, Bern University Hospital and University of Bern), Yvonne Pfeiffer (Swiss Patient Safety Foundation), Vanja Piezzi (Inselspital, Bern University Hospital and University of Bern), Luisa Salazar (Inselspital, Bern University Hospital and University of Bern), David Schwappach (Institute of Social and Preventive Medicine (ISPM), University of Bern, Switzerland), Benoit Sicre (Lucerne University of Applied Sciences and Arts), Bernard Surial (Inselspital, Bern University Hospital and University of Bern), Nicolas Troillet (Central Institute of Valais Hospitals), Andreas Widmer (University Hospital Basel), Marcel Zwahlen (ISPM, University of Bern).

**Collaborators** Swissnoso, National Center for Infection Control, Bern, Switzerland/ Members of Swissnoso (in alphabetical order): Carlo Balmelli, Lugano; Niccolò Buetti, Geneva; Delphine Berthod, Sion; Stephan Harbarth, Geneva; Philipp Jent, Bern; Jonas Marschall, Bern; Hugo Sax, Zurich; Matthias Schlegel, St Gallen; Alexander Schweiger, Zug; Laurence Senn, Lausanne; Rami Sommerstein, Lucerne; Nicolas Troillet, Sion; Sarah Tschudin Sutter, Basel; Danielle Vuichard Gysin, Frauenfeld; Andreas F Widmer, Basel; and Walter Zingg, Zurich.

**Contributors** YP—acquisition of data; analysis and interpretation of data; drafting of the manuscript; critical revision of the manuscript. AA—analysis and interpretation of data; drafting of the manuscript; critical revision of the manuscript. JMaag—acquisition of data; analysis and interpretation of data; critical revision of the manuscript. MAL—study conception and design; critical revision of the manuscript. DS—study conception and design; analysis and interpretation of data; drafting of the manuscript; critical revision of the manuscript. JMarschall—study conception and design; analysis and interpretation of data; critical revision of the manuscript. DS and JMarschall are responsible for the overall content as the guarantor.

**Funding** This work was supported by a research grant from the Swiss National Science Foundation (no: 179500; Project 'Understanding the drivers of surgical site infection: Investigating and modelling the Swissnoso surveillance data (Watussi)').

**Disclaimer** The funding source was not involved in study design, collection, analysis and interpretation of data gathered nor in the writing of or decision to submit the manuscript.

**Competing interests** None declared.

**Patient and public involvement** Patients and/or the public were involved in the design, or conduct, or reporting, or dissemination plans of this research. Refer to the Methods section for further details.

**Patient consent for publication** Not required.

**Ethics approval** This study involves human participants and was approved by the Cantonal Ethics Committee (Bern, project ID 2019-00294). Survey participation was considered informed consent.

**Provenance and peer review** Not commissioned; externally peer reviewed.

**Data availability statement** Data may be obtained from a third party and are not publicly available. Data supporting this study will be available in anonymised form for additional research under the specifications set forth by Swissnoso and the ANQ (Swiss National Association for Quality Development in Hospitals and Clinics). In principle, any researcher can submit a proposal to use Swissnoso data (including complementary data used in this study), which will be reviewed by the Swissnoso board.

**ORCID iDs**
Yvonne Pfeiffer http://orcid.org/0000-0001-6226-5037
David Schwappach http://orcid.org/0000-0001-8668-3065
Jonas Marschall http://orcid.org/0000-0002-0052-3210

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
