## [Reviewer comments · BMJ Open]

ARTICLE DETAILS

TITLE (PROVISIONAL)	Are cross-sectional safety climate survey results in operation room staff associated with the surgical site infection rates in Swiss hospitals?
AUTHORS	Pfeiffer, Yvonne; Atkinson, Andrew; Maag, Judith; Lane, Michael; Schwappach, David; Marschall, Jonas

VERSION 1 – REVIEW

REVIEWER	Saleem, Omema Dow University of Health Sciences
REVIEW RETURNED	23-Aug-2022

GENERAL COMMENTS	This article addresses the important safety issues of surgical patients. The rate of surgical site infection for a given category of surgical procedure depends on the adherence to the standard measures of health care safety. This study also explains the association of safety climate with surgical site infection along with some limitations that if addressed in future studies can help to improve the safety climate.
--

REVIEWER	Morikane, Keita Yamagata University Hospital, Infection Control and Clinical Laboratory
REVIEW RETURNED	21-Sep-2022

GENERAL COMMENTS	In this study, authors investigated the association between SSI incidence and safety culture & teamwork among hospital employees. Their relationship has not extensively investigated so far, and therefore is one of the interesting research area toward the better prevention of SSI. However, as authors state in the Discussion section, SSI is already known to be associated with various factors other than safety culture and teamwork. In this study, patient factors such as ASA score and duration of surgery seem to be adjusted. However, there are many more factors, including surgeons' technique and knowledge, laparoscope use for colorectal surgery, antimicrobial prophylaxis, which are known to affect SSI incidence. Safety climate would impact healthcare safety, but not necessarily closely associated with SSI incidence. Another concern is that this research involves several indicators including 'safety and teamwork' across 'climate and strength'. Authors extensively analyzed association of these factors with SSI incidence, but having too many variables which are interchangeably related would compromise the analyses. Minor comments: Appendix (page 45)
--

	The second title of the list is not 'SCS Safety Clinical Scale(SCS)', but should be 'Teamwork Climate (TWC)'. Table 1 (page 12) Adjusting infection rate in the NNIS system is based on the idea of stratifying surgeries using NNIS risk index and calculating Standardized Infection Ratio, not rate. The term 'NNIS adjusted infection rate' is not common. If authors are to use the term, it is necessary to explain how to calculate it.
--	---

REVIEWER	Fry, Donald Northwestern University, Surgery
REVIEW RETURNED	24-Oct-2022

GENERAL COMMENTS	This study examines the relationship between the climate of safety and teamwork in hospitals for the prevention of SSIs with the actual reported rates of these infections. The study failed to conclusively validate value to the climate of safety with lower SSIs with the possible exception of total joint replacement surgery. For colorectal and C-section there appeared to be a negative correlation. The authors have provided several explanations for the failure of the culture of safety to be translated into better results. One particular area that has not been emphasized in the discussion or the conclusions is the accuracy of the SSI surveillance data. The accuracy of reporting of SSIs is likely to be more thorough in those hospitals where results are being carefully scrutinized. In the era of shorter hospitalizations, more and more SSIs are not identified until after the patients are discharged from the hospital. Accordingly, hospitals with a climate of safety may be identifying more infections as a function of surveillance. The authors identify a bias as a potential issue in the identification of SSI in the bullet points at the beginning of the manuscript but they do not emphasize the point in the limitations section of the paper. This point must be emphasized. It is hard to believe that a culture of safety might have a negative effect upon the targeted outcome.
---

VERSION 1 – AUTHOR RESPONSE

Reviewer #1	Response
1. This article addresses the important safety issues of surgical patients. The rate of surgical site infection for a given category of surgical procedure depends on the adherence to the standard measures of health care safety. This study also explains the association of safety climate with surgical site infection along with some limitations that if addressed in future studies can help to improve the safety climate.	Thank you very much for the appreciation.

Reviewer #2	Response
1. In this study, authors investigated the association between SSI incidence and safety culture & teamwork among hospital employees. Their relationship has not extensively investigated so far, and therefore is one of the interesting	Thank you very much for this positive evaluation. We agree that the specific relation of safety climate with SSI rates needs more investigation than has previously been done.

research area toward the better prevention of SSI.	
2. However, as authors state in the Discussion section, SSI is already known to be associated with various factors other than safety culture and teamwork. In this study, patient factors such as ASA score and duration of surgery seem to be adjusted. However, there are many more factors, including surgeons' technique and knowledge, laparoscope use for colorectal surgery, antimicrobial prophylaxis, which are known to affect SSI incidence. Safety climate would impact healthcare safety, but not necessarily closely associated with SSI incidence.	Thank you, your points raised describe really well some of the issues the results of our study brought up and which we wanted to raise in the discussion section. In order to take up your thoughts and make them clearer in our discussion section, we made two changes.  - We added the term "safety climate" to the phrase on page 16: "First, safety climate surveys may not fully assess the perceptions that are actually relevant for performing specific safety behaviors such as adhering to SSI prevention measures." - We added the phrase on pages 13/14 in the discussion section: "This result may be interpreted in two ways: that a), safety climate is not as important as other factors that have not been assessed (such as knowledge of personnel or surgical technique used) or that b) safety climate was not assessed specific and well enough to uncover existing strong associations." Additionally, we now bring the results on climate strength earlier in the discussion section.
3. Another concern is that this research involves several indicators including 'safety and teamwork' across 'climate and strength'. Authors extensively analyzed association of these factors with SSI incidence, but having too many variables which are interchangeably related would compromise the analyses.	Thank you for bringing up this point. We agree that it is important to keep the number of variables in a manageable and interpretable size and also to keep track of potential confounding interconnections between variables. This was one of the reasons for which we decided to use the safety climate scale of the SAQ instead of the multi-dimensional instrument HSOPSC, for example. We added the teamwork dimension to assess potential variables that affect cooperation in the OR. As only investigating the level of safety climate and neglecting its strength has been reported to miss important drivers of the interaction between climate and its outcomes, it was important to include strength in the analyses. In order to better represent these rationales in the manuscript, we made the following changes:  - We slightly reworded the section explaining climate strength and the rationale for including it in the study and bring it now before stating the aims on page 4 of the manuscript.

	- We added “and for keeping potential intercorrelation between variables to a minimum” to the phrase in which we explain why we used a composite score for assessing safety climate, see page 5: “As prior research exploring the relationship between safety climate and SSI rates has advocated ⁹ and for keeping potential intercorrelations between variables to a minimum we used one composite safety climate scale.”
4. Minor comments: Appendix (page 45) The second title of the list is not ‘SCS Safety Clinical Scale(SCS)’, but should be ‘Teamwork Climate (TWC)’.	Thank you very much for pointing out this error. We corrected accordingly.
5. Table 1 (page 12) Adjusting infection rate in the NNIS system is based on the idea of stratifying surgeries using NNIS risk index and calculating Standardized Infection Ratio, not rate. The term ‘NNIS adjusted infection rate’ is not common. If authors are to use the term, it is necessary to explain how to calculate it.	We calculated the NNIS score on a patient level based on the components of the composite score for the specific surgery type (duration of surgery, ASA score, contamination class). We then calculated the hospital level weighted average SSI rate per year based on the patient NNIS scores. In this way, we enabled hospital level comparisons on a yearly basis adjusted for the components of the score.

Reviewer #3	Response
1. This study examines the relationship between the climate of safety and teamwork in hospitals for the prevention of SSIs with the actual reported rates of these infections. The study failed to conclusively validate value to the climate of safety with lower SSIs with the possible exception of total joint replacement surgery. For colorectal and C-section there appeared to be a negative correlation.	Thank you for your review and this condensed description.
2. The authors have provided several explanations for the failure of the culture of safety to be translated into better results. One particular area that has not been emphasized in the discussion or the conclusions is the accuracy of the SSI surveillance data. The accuracy of reporting of SSIs is likely to be more thorough in those hospitals where results are being carefully scrutinized. In the era of shorter hospitalizations, more and more SSIs are not identified until after the patients are discharged from the hospital. Accordingly, hospitals with	Thank you very much for raising this important point. We now discuss this limitation in the discussion section. On page 16, we added the following phrases: “The assessment of the surgical site infection rates may be biased due to underreporting of infections and thus conceal certain existing associations. As the surgical site infection rates also included an assessment of potential infections 30 days after the procedure in a telephone interview, this effect is minimized compared to studies assessing SSIs before discharge.”

a climate of safety may be identifying more infections as a function of surveillance. The authors identify a bias as a potential issue in the identification of SSI in the bullet points at the beginning of the manuscript but they do not emphasize the point in the limitations section of the paper. This point must be emphasized. It is hard to believe that a culture of safety might have a negative effect upon the targeted outcome.	
---	--

VERSION 2 – REVIEW

REVIEWER	Fry, Donald Northwestern University, Surgery
REVIEW RETURNED	03-Jan-2023
GENERAL COMMENTS	The revision is acceptable.